# The Beneficial Effects of Pine Nuts and Its Major Fatty Acid, Pinolenic Acid, on Inflammation and Metabolic Perturbations in Inflammatory Disorders

**DOI:** 10.3390/ijms24021171

**Published:** 2023-01-06

**Authors:** Rabaa Takala, Dipak P. Ramji, Ernest Choy

**Affiliations:** 1Division of Infection and Immunity, Tenovus Building, School of Medicine, Cardiff University, Cardiff CF14 4XN, UK; 2Cardiff School of Biosciences, Cardiff University, Sir Martin Evans Building, Museum Avenue, Cardiff CF10 3AX, UK; 3Department of Rheumatology, Heath Park, University Hospital of Wales, Cardiff CF14 4XW, UK

**Keywords:** inflammatory cytokines, polyunsaturated fatty acids, pinolenic acid, pine nuts oil, gene expression, microRNA, protein-coding genes, oxidative stress

## Abstract

Inflammatory disorders such as atherosclerosis, diabetes and rheumatoid arthritis are regulated by cytokines and other inflammatory mediators. Current treatments for these conditions are associated with significant side effects and do not completely suppress inflammation. The benefits of diet, especially the role of specific components, are poorly understood. Polyunsaturated fatty acids (PUFAs) have several beneficial health effects. The majority of studies on PUFAs have been on omega-3 fatty acids. This review will focus on a less studied fatty acid, pinolenic acid (PNLA) from pine nuts, which typically constitutes up to 20% of its total fatty acids. PNLA is emerging as a dietary PUFA and a promising supplement in the prevention of inflammatory disorders or as an alternative therapy. Some studies have shown the health implications of pine nuts oil (PNO) and PNLA in weight reduction, lipid-lowering and anti-diabetic actions as well as in suppression of cell invasiveness and motility in cancer. However, few reviews have specifically focused on the biological and anti-inflammatory effects of PNLA. Furthermore, in recent bioinformatic studies on human samples, the expression of many mRNAs and microRNAs was regulated by PNLA indicating potential transcriptional and post-transcriptional regulation of inflammatory and metabolic processes. The aim of this review is to summarize, highlight, and evaluate research findings on PNO and PNLA in relation to potential anti-inflammatory benefits and beneficial metabolic changes. In this context, the focus of the review is on the potential actions of PNLA on inflammation along with modulation of lipid metabolism and oxidative stress based on data from both in vitro and in vivo experiments, and human findings, including gene expression analysis.

## 1. Introduction

Fatty acids (FAs) are widespread in the environment and are the main constituents of triacylglycerols (TAGs), phospholipids (PL) and other lipid compounds [1]. They are important in biological and metabolic functions as they are key components of cellular organelles. Composition of cell membrane FA is affected by genetics, diet, metabolism, and hormones [1]. The physical and functional characteristics of cellular membranes, including signaling and traffic within the membrane, are affected by the amount of FAs present in the membrane. Polyunsaturated FAs (PUFAs) are widely present in plants and marine sources. Hitherto, having demonstrated that many plant-derived FAs have inflammatory suppressing effects [1], it could be a good alternative to marine source FAs with diminishing fish stocks. So more sustainable sources of health-promoting FAs need to be explored. Thus, Pinolenic acid (PNLA) as a plant-derived FA may be a candidate and it has become popular in recent years due to its ability to suppress appetite and reduce weight [2]. PNLA is a Δ5-unsaturated polymethylene-interrupted FA (UPIFA) (Figure 1). Its structure is analogous to that of linoleic acid (LA) (n-6 and γ-linolenic acid (GLA) precursor); GLA (n-6 and dihomo-γ-linolenic acid (DGLA) precursor), and α-linolenic acid (ALA; n-3 and the originator of both docosahexaenoic acid (DHA) and eicosapentaenoic acid (EPA)) [3] (Figure 1).

Some PUFAs such as PNLA are examples of nutraceuticals which are believed to have anti-inflammatory effects through intracellular signals that activate anti-inflammatory pathways, such as peroxisome proliferator-activated receptors (PPARs) [4,5,6], and suppress pro-inflammatory pathways/mediators such as nuclear factor kappa-light-chain-enhancer of activated B cells (NF-κB) [5,6], arachidonic acid (AA)-derived eicosanoids, and reactive oxygen species (ROS) generation with generally an excellent safety profile [3,4,5,6]. Amr and Abeer showed that pine nuts oil (PNO) possesses a wide therapeutic effect, such as antibacterial, antifungal, antiviral, antiseptic, expectorant, cholagogue, choleretic, diuretic, and antihypertensive [7]. In human studies, PNLA has been demonstrated to have anti-inflammatory, immune-metabolic, anti-rheumatic and anti-atherogenic actions [5,6] together with weight-lowering properties [2]. According to many studies, PNO and PNLA have the potential to be used as anti-cholesteric agents [3,7,8]. In this review, we summarise the evidence suggesting that PNLA can work as an anti-inflammatory agent. 

## 2. Composition of PNLA and Pine Nuts Oils

Pine nuts and PNO consumption have increased over the last few decades. The biggest distributors are in Asia, including China, North Korea and Pakistan. Korea, the USA, and Russia are the biggest users of pine nuts and PNO [9]. Pine nuts have a unique flavour with very high caloric and lipid content. Pine nuts are comprised of almost 64% fat, 20% protein and 5% carbohydrate [10,11]. Most of the FAs in pine nuts are unsaturated FAs (UFAs) with only 10% saturated FAs (SFAs) [12]. The pine nuts are 9–12 mm long [13] and have a high oil content (45–65 g/100 g of nuts) [12,13]. PNLA is an unusual and unique PUFA that is found only in PNO such as *Pinus orientalis* and maritime pine (*P. pinaster*) seeds, which are rich sources of PNLA [9,13]. Pine nuts from *P. koraiensis* (Korean pine), *P. sibirica* (Siberian pine), *P. gerardiana* (chilgoza pine) and *P. pinea* (stone pine) are the most common genera eaten by consumers [9]. The nut oils from *P. eldarica*, *P. excelsa*, *P. pinea* and *P. torreyana* have low level of PNLA (very low in the case of *P. pinea*) and are rich in LA or oleic acid [14]. PNO also contains antioxidant lipid-soluble components, including tocopherols, phytosterols and squalene.

The FA composition of pine nuts (*P. sibirica* and *P. koraiensis* oils) has been defined and found that LA (all-*cis*-9-, -12–18:2) is the major and most abundant PUFA (48.4%) in PNO. Oleic Acid (*cis*-9-18:1) is the second most common monounsaturated FA (MUFA) (24%). PNLA is the most common UPIFA, typically containing around 15 to 20% of total FAs and taxoleic acid represents about 1.8% [3]. The two main SFAs in PNO are palmitic acid (16:0) and stearic acid (18:0). Korean pine nuts (KPN) and Siberian pine nuts (SPN) are rich sources of PNLA which forms up to 20% and 27%, respectively, of their total content.

## 3. PNLA and Its Metabolism

In mammalian systems, the metabolism of PNLA has not been well studied. It was reported that PNLA is not converted to AA but can reduce AA levels in the phosphatidylinositol fraction of HepG2 cells from 16% to 8.7% [15]. PNLA is quickly incorporated, metabolized, and elongated to Δ-7 eicosatrienoic acid (7 ETA; all *cis*-7, -11, -14-20:3) by the FA elongation system. After PNLA treatment, a high proportion (up to 30%) of ETA is found in a variety of cell membranes PLs [16,17,18,19,20,21,22]. A small proportion of this elongated metabolite is further elongated to form the (*cis*-9,-13,-16–22:3) form. The amount of PNLA and its metabolites that are incorporated into the cell membrane is affected by the concentration of PNLA and the duration of incubation [17,19,20]. For example, the level of Δ-9,-13,-16–22:3 form in RAW264.7 macrophages increased as the quantity of PNLA in the culture medium and the incubation time increased [16]. PNLA altered the composition of FAs in PL in murine RAW264.7 macrophages, with reduction in the levels of 18- and 20-carbon PUFA and an increase in 22-carbon FA [16]. This means that the levels of SFA decrease while UFA increases in PL fraction; this might be one reason for the anti-inflammatory actions of PNLA. Figure 2 shows the metabolic pathway of PNLA.

Elongase 5, an enzyme encoded by the Elongase of Very-Long Fatty Acid 5 (ELOVL5) gene, elongates mono- and poly-unsaturated FAs of 18–20 carbons in length. Elongase 5 catalyses the long-chain FA elongation cycle and is the initial and rate-limiting step. It is also responsible for the elongation of PNLA as demonstrated and shown recently [20]. Most of the anti-inflammatory effects such as reduction in the generation of inflammatory mediators, including interleukin (IL)-1β, IL-6, tumour necrosis factor (TNF)-α and prostaglandin (PG)E2, in the EA. hy296 endothelial cell line following PNLA incubation were attenuated by silencing elongase 5 implicating that PNLA acts via its elongation product [20]. 

Tanaka et al. in 2007 studied the metabolic pathway of some conifer oil-derived PUFAs (sciadonic and juniperonic acid) which were metabolized to essential FAs in animal cells; the accumulated LA was elongated in rat liver microsomes and it was suggested that sciadonic acid underwent β-oxidisation pathway [23]. Likewise, in vitro studies based on rat liver microsomes detected the C2-elongated metabolite of PNLA, although it has not been found in in vivo experiments [18]. A potential reason for these contrasting observations is that in vivo, the C2-elongated metabolite of PNLA might be rapidly translocated to the mitochondria and undergo β-oxidation. Most dietary linoleate and K-linolenate in rats were β-oxidised with only 3.0% and 1.4% converted to longer PUFAs, respectively [18]. Because PNLA is a substrate of carnitine-palmitoyl transferase I than K-linolenate, practically all the PNLA in pine seed oil (PSO) was oxidized by the mitochondrial β-oxidation pathway, hence the C2 chain-elongated metabolite of PNLA possibly may not get to the level seen in in vitro experiments [18]. 

## 4. PNO and PNLA Inhibit the Inflammatory Response

The anti-inflammatory effects of PNO have been shown in in vitro and in in vivo animal models [3,4,10,24], which have also been demonstrated with PNLA [5,6,15,16,17,19,25]. Dietary PUFAs impact inflammation by several mechanisms, including altering membrane function and structure, and regulating the synthesis of lipid mediators [1,26]. Recently, Takala et al. have shown that the anti-inflammatory effects of PNLA are associated with alternations in gene expression and intracellular signalling networks [5,6] with the suppression of NF-κB [5,6,20,26] and signal transducer and activator of transcription (STATs) being key potential mechanisms [5,6]. Such transcription factors could potentially regulate enzymes, including cyclooxygenase-2 (COX-2) and inducible nitric oxide synthase (iNOS) as well as proinflammatory cytokines such as TNF-α and IL-6. Moreover, bioinformatic analysis [5] suggested that anti-inflammatory transcription factors, including PPARs [4,5,6,27,28] and its co-activator PPAR-coactivator 1-alpha (PGC-1α), would be stimulated. PGC-1α is the product of the PPARGC1A gene. PPARs inhibits NF-κB, STAT1, activator protein 1 (AP-1), and STAT3. In addition, PNO and PNLA inhibit the production of NO [3,17], PGE2 [5,16,17] and pro-inflammatory cytokines [3,5,17,19]. The overall anti-inflammatory effects of PNLA and PNO’s are summarized in Table 1. 

### 4.1. Effects on Cell Culture

*E. coli* lipopolysaccharide (LPS) stimulated murine microglial BV-2 cells and incubated with PNLA at a concentration of 50 µM, led to a reduction in the synthesis of nitric oxide (NO), IL-6 and TNF-α by 27–74% [17]. PNLA also significantly reduced NO and PGE2 levels by 35% in rat primary peritoneal macrophages following LPS stimulation [29]. In another study in HepG2 cells, PNLA (25 μM) decreased NO produced by 50% [25]. Chen et al. assessed the effects of PNLA in LPS stimulated THP-1 macrophages and found that the levels of IL-6, TNF-α, and PGE2 were reduced by 46%, 18%, and 87%, respectively [19]. Previously, PNLA has been shown to reduce PGE2 synthesis in LPS-stimulated murine RAW264.7 macrophages [16]. In both studies, the effects of PNLA were dose dependent. PNLA also reduced PGE1 or PGE2 generation in RAW264.7 macrophages stimulated with DGLA or AA, by 53% and 22%, respectively [16].

When RAW264.7 macrophages are stimulated with LPS, iNOS and COX-2 mRNA and protein expression are increased, which then increases the production of NO and PGE2. The NF-κB pathway is normally activated during the up-regulation of the iNOS and COX-2 genes. Chen et al. (2015) demonstrated that PNLA reduced the expression of iNOS and COX-2 caused by LPS by 55% and 10%, respectively [17]. This shows that, as has been demonstrated for n-3 PUFAs, PNLA may block the activation of NF-κB [1,28]. Huang et al. reported similar results using 50 μM PNLA treatment which reduced COX-2 and PGE2 release from LPS-stimulated RAW264.7 and rat primary peritoneal macrophages [29]. However, Chuang et al. [16] found an increase in COX-2 protein expression (12%) in murine macrophages, despite lowered PGE2 production. They suggested that the decrease in PGE2 production may be due to the competition of PNLA or its metabolites with AA as a substrate for COX-2 [16]. Consistently with findings in LPS stimulated RAW264.7 cells, PNLA prevented NF-κB activation [29], and Baker et al., demonstrated that PNLA (50 μM) decreased TNF-α stimulated NF-κB activity in EA.hy296 cells [20]. Additionally, ICAM-1, MCP-1, regulated on activation, normal T cell expressed and secreted (RANTES) production by TNF-stimulated EA.hy296 cells were all lowered by PNLA treatment, as well as adhesion to human THP-1 monocytes [20,27].

One study reported that 50 and 100 μM PNLA reduced PGE2 generation by 12-*O*-tetradecanoylphorbol-13-acetate (TPA)-stimulated human MDA-MB-231 breast cancer cells with lower COX-2 mRNA and protein levels [22]. Further assays in these cells showed that n-3 (DHA) or PNLA reduced cell invasion by 30% and 25%, respectively, and both PNLA and DHA inhibited cell motility [22]. The experiments demonstrated that PNLA was the most potent of the FAs (PNLA, DHA or EPA) in reducing PGE2 production. PNLA reduction of TPA induced PGE2 production was dose-dependent [22]. Reduction of PGE2 by PNLA, DHA, or EPA appeared to be mediated in part by decreased COX-2 expression [22].

### 4.2. Effects on Animal Models

Several research on animals has also highlighted PNLA’s anti-inflammatory properties. PNLA reduced the release of IL-1β, IL-6, TNF-α, and PGE2 in the TPA-stimulated dorsal skin of a mouse model where PNLA or vehicle was applied topically to the shaved back skin. This was linked to a decrease in phosphorylation of p38- and c-Jun N-terminal kinase (JNK)-mitogen-activated protein kinase (MAPK), but not of extracellular signal-regulated kinase-MAPK [19]. Supernatants from dorsal skin tissue homogenates were also assessed [19]. Leukocyte, neutrophil, and macrophage infiltration were reported to be decreased after a single injection of PNLA (3 g) in mice with TPA-induced ear swelling [19]. The authors suggested that these effects may result from the direct regulation of cell signalling rather than the uptake of PNLA into the cells. 

In rats, PNLA administered orally prior to carrageenan injection into the right foot reduced oedema [24]. PNLA applied topically to the feet also reduced fever. Additionally, after PNO was injected into the right hind paw, the reaction time to a hot plate increased by 1.4-fold [24]. This shows that COX-2 activity and PGE2 release may play a role in PNLA’s analgesic effects. 

The effects of dietary PNO on immune function were examined in other animal studies. Rats were given *P. koraiensis* oil by Matsuo et al. and then intraperitoneal ovalbumin as an immunization [30]. Rats given safflower oil (a source of LA; n-6) or evening primrose oil (EPO; a source of GLA; n-6) had reduced CD4 T-cell counts, as well as spleen cells’ synthesis of Leukotriene B4 (LTB4) and immunoglobulins (Ig)-E and-G, compared to those given PNO [30] suggesting that PNLA may modulate the immune response. Park et al. found that PNO feeding led to an increase in concanavalin A-stimulated splenic lymphocyte proliferation and IL-1β production by splenocytes activated with LPS [31]. Lin et al. (2017) also reported that a low dose of *P. koraiensis* could enhance the immune function in vivo, with elevated quantities of IL-2, IL-4, IL-10, and interferon (IFN)-γ in mice [32]. These observations contradicted the reported anti-inflammatory effects of PNLA [16,17]. However, these immune-enhancing effects may be due to non-PNLA components in PNO or inherent differences in the models. 

### 4.3. Effects on Healthy Individuals and Patients with Chronic Inflammatory Diseases

More recently, Takala et al. confirmed some of the anti-inflammatory actions of PNLA in healthy volunteers and RA patients [5]. In PBMCs from RA patients stimulated with LPS, PNLA decreased IL-6 and TNF-α release by 60%, whereas in HCs, it did so by 50% and 35%, respectively. LPS-induced PGE2 levels in such PBMCs from RA patients and HCs were also reduced substantially by PNLA. Regarding IL-1β, levels were reduced in supernatants of activated PBMCs of HCs while unaffected in RA patients after treatment with PNLA. 

When intracellular levels of IL-6, TNF-α, IL-1β and IL-8 in CD14 monocytes isolated from active patients with RA were assessed, there was a reduction in TNF-α, IL-6 and IL-1β, all approximately by 25%, and reduced IL-8 level by 20% without an effect on MCP-1 expression [6]. PNLA also significantly reduced the proportion of LPS-activated CD14 monocytes in RA patients from 66.8% to 58.4% and 56.3% for 25 and 50 μM PNLA, respectively [6]. There was no correlation between the reduction in different cytokines expressing CD14 monocytes by PNLA and the clinical and laboratory features of RA or disease activity scoring.

Takala et al. also used whole genome transcriptomics and investigated both inflammatory models of PBMCs from RA patients and healthy volunteers, and purified CD14 monocytes from active RA patients that were activated with 100 ng/mL LPS following pre-incubation with 25 μM PNLA. The bioinformatic analysis showed that NF-κB, STATs and chemokine receptor 2 (CCR2) were inhibited relative to the same model which was vehicle-treated and LPS stimulated, while up-regulated expression of PPARs was observed [5,6]. PNLA also regulated the expression of metabolic genes, including pyruvate dehydrogenase kinase-4 (PDK4) and serpin family E member 1 (SERPINE1) that codes for plasminogen activator inhibitor-1 (PAI-1) from HCs, and fructose-bisphosphatase 1(FBP1), PDK4 and N-Myc downstream regulator 2 (NDRG2) from RA patients [5] as shown in Table 2 and Table 3, respectively. In addition, in 2022, they reported that there are many protein-coding genes whose expression was affected by 25 μM PNLA treatment of purified CD14 monocytes from patients with RA stimulated with 0.1 μg/mL of LPS as shown in Table 4. These studies also demonstrated that PNLA is involved in NF-κB activity inhibition and thereby subsequent downstream effects on inflammatory mediators. In this regard, PNLA has similar effects to n-3 PUFAs (EPA and DHA) which were reviewed in [1,26]. 

## 5. PNO and PNLA Inhibit Oxidative Reactions 

Dietary antioxidants may improve chronic diseases by reducing oxidative damage [33]. Antioxidant enzymes are essential for cell defense against the harm that free radicals may do to cells and macromolecules. Two of these enzymes are phospholipid hydroperoxide glutathione peroxidase (GSH-Px) and superoxide dismutase (SOD). *P. koraiensis* oil was included in a meal given to rats, and they showed elevated blood levels of SOD and GSH-Px activity [34]. Malondialdehyde (MDA) is a sign of lipid degradation caused by free radicals and is commonly used as a biomarker for measuring lipid peroxidation in cell membranes. There was a reduction in serum MDA levels in the PNO-fed group relative to the control [34]. Additionally, in comparison to the controls, the pine high-fat diet (PHFD) group had the greatest level of SOD2 mRNA compared to PNO and soybean oil (SBO)-fed groups and soybean high-fat diet (SHFD) groups [34], which might shield from damage due to ROS. In H_2_O_2_-induced HepG2 cells, 1, 5, and 10 μM of PNLA, dose-dependently reduced intracellular ROS accumulation by 29.1, 59.1, and 65.5% and intracellular MDA content by 15.96%, 20.5%, and 22.9%, respectively [35]. These data suggest that PNLA reduces ROS-induced lipid peroxidation, thereby decreasing oxidative damage [3]. Although Takala et al. found that PNLA (25–100 μM) has no effect on ROS in THP-1 monocytes and macrophages in vitro [5], they found that 25 μM PNLA reduced NO production by PBMCs (unpublished observation). NO is considered one of the oxidative stress mediators. 

Nuclear factor erythroid 2-related factor 2 (NRF2) was translocated to the nucleus by PNLA in order to protect cells from oxidative stress, and the expression of the antioxidant enzymes heme oxygenase-1 (HO-1) and quinone dehydrogenase-1 that are downstream of NRF2 was also increased [35]. The NRF2 transcription factor is a key regulator in cellular antioxidant defense responses [36]. The Kelch-like ECH-associated protein 1 (KEAP1) expression was decreased; according to Zhao et al., these changes also activate the production of antioxidant enzymes leading to a decrease in MDA production, which protects against damage by ROS [35]. 

Oxidative damage and repair are perpetual processes in vivo. Inflammation is exacerbated by oxidative damage, which is increased when damaged biomolecules are not replaced [37,38]. Under inflammatory conditions, as shown in Figure 3, hypoxia inducible factor-1 (HIF-1), a member of the basic ARNT-protein family and typically a heterodimer consisting of HIF-1α and HIF-1β subunits, provides a means of regulating gene expression that operates at a molecular level in response to the availability of oxygen. This idea inspired the production of medications that have been licensed for inhibiting HIF-1 [39]. HIF-1 activates pro-inflammatory genes such as NF-κB and STATs. STATs, once phosphorylated by activated Janus Kinases (JAKs), enables the production of cytokines that are pro-inflammatory. A transcription factor called Aryl hydrocarbon receptor nuclear translocator 2 (ARNT2) controls the transcription of HIF-1. ARNT2 expression was found to be inhibited by 25 μM PNLA in PBMCs from RA patients (unpublished data). Takala et al. also found that oxidative phosphorylation (OXPHOS) is the most significant canonical pathway to be inhibited in purified CD14 monocytes isolated from active RA patients followed by mitochondrial dysfunction [6]. PNLA potentially reduces lactate dehydrogenase (LDH) via activation of PDK4, which leads to less lactate and acetyl CoA that consequently results in the inhibition of OXPHOS [6]. By inhibiting OXPHOS, PNLA may reduce the ability of pathogenic cells to generate sufficient ATP to survive, proliferate, and function, thus promoting the resolution of inflammation through metabolic reprogramming. Some of the putative molecular activities of PNLA based on the control of OXPHOS are summarized in Figure 3. It may be possible to gain insight into the interaction between inflammation, immunological metabolism, and some antioxidant protective activities of PNLA by learning more about the factors that affect the way that PNLA functions in chronic inflammatory disorders.

## 6. PNLA and PNO protect against Hyperlipidaemia and Atherosclerosis and Regulate the Lipid and Metabolic State Based on Cell Line, Animal, and Human Studies

The imbalance between exercise and dietary intake is considered the main cause of an increase in body fat [40,41]. An increase in body fat, especially visceral fat, is associated with diseases, including osteoarthritis, cardiovascular disease (CVD), hyperlipidaemia, hypertension, and type 2 diabetes (2DM) [42,43]. Weight reduction and reducing visceral fat accumulation is therefore important for health.

Pine nuts are cholesterol-free and a good source of nutrients. According to the outcomes of studies on animals, PSO considerably reduced blood pressure and had cholesterol-lowering effects [44,45]. PNO diminished lipid accumulation in the rat liver and produced a decrease in lipid profile, lipoprotein cholesterol, and liver weight, which was concentration-dependent [7]. It had no hepatotoxic or detrimental effect on the liver as hepatic enzymes, such as serum alanine transferase (ALT), aspartate transferase (AST) and alkaline phosphatase (ALP) levels, were unaffected by PNLA in rabbits [8]. Jin et al. also reported that even though there was a trend of increasing AST, ALT, protein urea nitrogen, and creatinine in animals fed with high fat diet (HFD), the levels remained normal showing that PNO was neither hepatotoxic nor nephrotoxic in mice [46]. 

Dietary maritime pine seed oil (MPSO) has been demonstrated to reduce TG, VLDL-triglycerides, and VLDL-cholesterol in rats in vivo, and diminished cholesterol efflux in vitro in cultured cells of mice expressing human apolipoprotein (Apo) A-I, according to a study by Asset et al. [45]. Moreover, a diet containing MPSO reduced HDL and ApoA-I levels when human ApoA-I and ApoB genes are expressed in transgenic mice. There are several hypotheses that have been suggested for hypolipidemic properties, one of them is that PNLA alters the expression of various Apo genes that have an essential role in lipid metabolism. Key proteins of HDL are ApoA-I and ApoA-II, lipoprotein remnant clearance requires ApoE, and the main component of TG-rich bioparticles is ApoC-III, which reduces VLDL lipolysis and uptake [47]. Other mechanisms that have been suggested for TAG-reducing properties of PNLA are diminished substrate availability to produce lipoproteins, lowered de novo lipid synthesis, or alterations in very low-density lipoprotein (VLDL) physicochemical properties [45,47]. Furthermore, PNLA reduced the activity of the lipid anabolic pathway in vitro in hepatoma HepG2 cells by downregulating the expression of genes associated with lipid synthesis and lipoprotein uptake. For example, long-chain acyl-CoA synthetases (ACSLs) mediates the formation of fatty acyl-CoA from FAs (lipid biosynthesis). In comparison to the BSA control group, PNLA decreased the mRNA levels of ACSL3, and ACSL4 by 30 and 20%, respectively [8]. 

Many inflammatory processes are generally initiated by monocyte migration; in the case of atherosclerosis, monocytes migrate from the lumen into the intima, and in RA they migrate into inflamed synovium. Monocytes once migrated undergo pathologic transformation into macrophages and commence the inflammatory process within the artery or the synovial joint. Recently, Takala et al. demonstrated that PNLA reduced migration of THP-1 monocytes induced by MCP-1 by more than a half in vitro. Macropinocytosis and DiI-oxLDL uptake were also decreased by 50% and 40% in differentiated THP-1 macrophages and by 40% and 25% in HMDM, respectively [5]. Both processes lead to lipid accumulation and the formation of atherogenic plaque.

PNO-FA induced higher internalization of Dil-LDL in HepG2 cells [48] suggesting that PNO-FA may lower LDL by increasing uptake by the liver. When compared to other oils, rats fed PNO had lower serum TAG and VLDL-TAG concentrations, according to Asset et al; however, despite the effects being relatively high, the differences were not statistically significant [45]. Lower serum total TAG levels in rats fed *P. koraiensis* oil compared to rats fed lard were similar but significant in findings reported by Park et al. [11].

According to Park et al. (2016), the PHFD group’s response on fatty acid synthase (FAS) mRNA expression was considerably less than that of the SHFD group. Lipogenesis was decreased in the PHFD group but not in the SHFD group [49]. After 16 weeks of feeding, Ferramosca et al. (2008) found that PNO-fed mice had considerably lower activity levels of hepatic lipogenic enzymes such as acetyl-CoA carboxylase and FAS and that PNO-fed mice were protected from liver fat accumulation when fed conjugated linoleic acid (CLA) [50,51]. Compared to SHFD, the body weight gain and mass of white adipose tissue were lower in PHFD (10% and 18% lower, respectively). In comparison to the SHFD, the PHFD had much lower liver TG levels (26% lower). Intake of PNO increased hepatic acyl-CoA dehydrogenase long chain (ACADL) mRNA levels [51]. These data indicate that PNO increases hepatic FA-β-oxidation and reduces lipogenesis.

PNLA was predicted to activate PPARs in RA patients and in healthy volunteers [5,6]. PPAR-γ increased cholesterol efflux from macrophages by the hepatic liver X receptor pathway [52,53]. In addition, PPAR-γ regulates adipocyte differentiation and glucose homeostasis as well as intestinal homeostasis through modulation of NF-κB-mediated pro-inflammatory responses [52,53,54]. Low PPAR-γ reduces the storage of fat in adipose tissue resulting in fat deposits in non-adipose tissues. PPAR-γ also regulates plasma leptin levels, insulin sensitivity, glucose homeostasis and blood pressure as well as the effects of antidiabetic drugs. Additionally, it prevents the release of inflammatory cytokines and the activation of macrophages [52]. Atherosclerosis in animal models and human studies are inhibited by PPAR-agonists [52,55,56], so they are potential treatments for CVD and 2DM [57,58]. All of these suggest that there is a metabolic beneficial action of PNLA which can be mediated through PPAR-γ activation although there is a controversy regarding the PNO effect that may downregulate PPAR-γ as reported by [46].

Compared to Safflower oil (SAO), *Pinus koraiensis* oil lowered blood cholesterol levels and decreased age-associated increases in blood pressure in rats, although Flaxseed oil (FSO) was more effective than PNO in this study [44]. PNO stimulated aortic PGI production more than SAO, and the difference between PNO and SAO groups was significant. PNO therapy also prevented ageing-related rise in blood pressure after 5 weeks and was sustained through 8 weeks [44]. The authors concluded that pine nut extracts not only reduce blood pressure but decrease damage to kidneys caused by hypertension. Pine bark extracts and pine nuts have also shown promising cardiovascular (CV) benefits, including improvement in endothelial function, decreasing plasma fibrinogen, and reducing plasma viscosity and systolic blood pressure [3,27,59,60]. These results have been attributed to the different types of FAs in pine nuts, especially PNLA or some of its metabolites. In a study by Amr and Abeer, rats fed a high-fat diet developed thickened tunica media associated with the proliferation of smooth muscles in the aorta, in addition to vaculations of tunica media in the aorta. In the second group of rats, cholesterol diet supplemented with 5 and 10% pine nut had vacuolations of tunica media. However, in the third group, rats fed a diet supplemented with 15% pine nuts showed normal histological structure [7].

When compared to maize oil in an acute murine glucose tolerance test, PNO and pure PNLA (free and esterified) substantially improved glucose tolerance [61]. In vitro results indicated that PNLA is a potent dual free FA receptor-1 and free FA receptor-4 (FFA1/FFA4) agonist [61,62] that has anti-diabetic effects. Different types of medium-to long-chain non-esterified FA (NEFA) have been linked in diverse ways in 2DM and other metabolic and inflammatory illnesses, and FFA1 and FFA4 are G protein coupled-transmembrane receptors that are activated by these NEFA. FFA1 is expressed in pancreatic β cells while FFA4 is expressed in intestinal entero-endocrine cells, macrophages, the pancreas, fat tissue, and the brain. It seems that co-activation of FFA1 enhances insulin production that is glucose-dependent [63,64], and FFA4 promotes insulin sensitivity and has anti-inflammatory effects [65,66]. The ethylene-interrupted PNLA was among the most effective NEFA on FFA1 and FFA4 in comparison with ALA, AA, DGLA, and GLA. PNLA outperformed all these FAs in promoting health and improved insulin sensitivity [61].

In various studies where mice are given HFDs containing PNO, there was weight reduction and less weight gain [31,49]. White adipose tissue mass was decreased because of these alterations [31,49,67], most likely as a result of increased oxidative metabolism and thermogenesis, which drives the utilization of fuel sources and reduces lipid accumulations. Therefore, the effect of PNO and PNLA on weight could be the result of both increased energy expenditure and decreased intake. Earlier in 2008, Hughes et al. reported a rise in the satiety hormones glucagon-like peptides (GLP-1) and cholecystokinin (CCK-8), in postmenopausal obese women participating in a pilot double-blind placebo-controlled clinical trial [2,68] when taking 3 g Korean pine nuts (KPN)-FFA in comparison with 3 g KPN-TG or 3 g olive oil. The levels of circulating CCK-8 in the PNO FFA and PNO TAG groups, respectively, were 60% and 22% higher. Comparing the PNO FFA group to the control, GLP-1 rose by 25% [2, 69]. Food intake also diminished by 36% in the PNO FFA group [2,68,69] suggesting that pine nuts may also be an appetite suppressant. 

## 7. Novel Potential Metabolic, Anti-Inflammatory, and Immune-Regulatory Effects Discovered by Transcriptomic and Bioinformatic Analyses

In C57BL/6 mice, Zhu et al. discovered decreased hepatic mRNA expression of ACADL, adipose triglyceride lipase (ATGL), carnitine palmitoyltransferase (CPT) 1A, and Apo-B100 as well as decreased gut mRNA expression of both CD36 and Apo-A4 [67]. This implies that dietary PNO may reduce the uptake of intestinal FA and increase hepatic mitochondrial FA oxidation. 

PNLA downregulated the expression of genes related to FA biosynthesis (fatty acid synthase (FAS), sterol regulatory element-binding protein 1(SREBP1), and stearoyl-CoA desaturase 1 (SCD1)) in the human hepatoma cell line (HepG2) when compared to control (53, 54, and 38% lower, respectively). Furthermore, the mRNA levels of genes related to cholesterol synthesis and lipoprotein uptake (3-hydroxy-3-methyl-glutaryl-coenzyme A reductase (HMGCR), and LDL receptor (LDLr), respectively) were significantly suppressed (30, and 43%, respectively) by PNLA [8,27]. The cholesterol-reducing capability of PNLA has been associated with a greater hepatic accumulation of PNLA and regulation of hepatic activity through boosted hepatic LDLr gene expression and elevated levels of SREBP2 [70].

Studies of protein–protein interaction networks (PPINs) by bioinformatic analyses are crucial for delineating how intracellular proteins work. Investigating a protein’s interaction with other proteins whose activities are known can help reveal those proteins whose functions are not fully identified [46]. Therefore, PPIN analysis was conducted to shed light on the molecular mechanisms underlying PNLA’s metabolic and immunological effects as demonstrated by Takala et al. [5,6] and Jin et al [46]. Takala et al. identified that 25 μM PNLA treatment of LPS-activated CD14 monocytes activated Kruppel-like factor 15 (KLF15), argonoute-2 (AGO2), ficolin (FLCN), Sirtuin3 (SIRT3) and Sirtuin signalling pathways and reduced the activity of STAT3, Transcription factor E3 (TFE3), and Death associated protein-3 (DAP3) pathways [5,6]. 

KLF15 is a transcription factor that controls inflammation and other pathogenic processes connected to atherosclerosis, as well as control of different signalling pathways [71]. It modulates the acetylation status and the activity of NF-κB and controls the activation of vascular smooth muscle cells (VSMCs) [72]. Several CVDs, including heart failure, progressive inflammatory vasculopathy and aortic aneurysm, have been linked to decreased KLF15 expression [71]. KLF15 mRNA expression is decreased in human aortic atherosclerotic tissue compared with nonatherosclerotic control aortae [71]. Overexpression of KLF15 in the human EA.hy926 cells exhibited a protective effect against TNF-α induced dysfunction and downregulated the levels of phospho-p65 (p-p65), ICAM-1, MCP-1, and TGF-β [71]. AGO2 plays a role in RNA interference and the silencing of miRNA is influenced by Ago proteins [73,74,75]. PNLA actions might be mediated in part through its effect on AGO2, and thereby pro-inflammatory miRNAs. In activated macrophages, TFEB and TFE3 control autophagy and lysosomal activity and are crucial for the transcription of pro-inflammatory cytokines such TNF-α and IL-1β [76]. DAP3 is a small subunit protein found in the mitochondrial ribosome involved in mitochondrial function, and various forms of cell death [77]. Overexpression of DAP3 leads to increases in cell death and osteoclastogenesis [77]. DAP3 acts downstream of pro-apoptotic cytokine stimuli such as IFN-γ and upstream of several caspases, in particular caspases 8 and 9 [77]. Some of the PNLA actions might be mediated through its inhibitory effect on DAP3 such as apoptosis and the inhibitory effect of these pro-inflammatory cytokines. In PBMCs from healthy volunteers, PNLA dramatically increased the expression of the PAI-1, which produces a protein that inhibits the enzymes responsible for forming plasmin from plasminogen, tissue plasminogen activator (tPA) and uro-plasminogen activator (uPA). In RA synovium, uPA expression is high in the synovial lining layers [78]. In TNF-transgenic mice, lack of plasminogen lowers synovial inflammation and joint degeneration through decrease in cytokines and matrix metalloproteinases (MMPs) [79]. 

FBP1 is part of the glycolytic pathway through which fructose 6-phosphate is phosphorylated. The expression of gene for FBP was also increased by PNLA in PBMCs from RA patients. It has anti-inflammatory properties, and the administration of FBP reduced arthritis with decreased joint swelling and pro-inflammatory cytokine levels [80].

The expression of ACADL and PPAR-α, γ and δ mRNAs were upregulated in human PBMCs and CD14 monocytes [5,6]. PNLA potentially elicited ligand activity for PPAR-α and PPAR-δ as treatment with this FA upregulated the downstream target genes of PPAR-α and PPAR-δ involved in FA oxidation, including peroxisome proliferator-activated receptor-gamma coactivator (PGC)-1α, mitochondrial uncoupling protein 3 (UCP3), carnitine palmitoyl-transferase 1B (CPT1b), acyl-CoA dehydrogenase medium chain (ACADM), and ACADL in the C2C12 myotubes cell line [34] and PBMCs from HCs and RA patients [5]. Park et al. have suggested that the ligand activity of PNLA for PPAR-α and PPAR-δ might be the main mechanism for the reduced adiposity in PNO-fed mice [31,49]. This activation was also suggested by others as one of the reasons for the anti-inflammatory properties of PNLA such as inhibition of AP-1, NF-κB and STATs [5,27,81]. In order to determine whether PNO affected the levels of PPAR-γ and SREBP-1c expression in the epididymal fat tissue, Jin et al. used RT-PCR. While orlistat, a weight-loss medication, and PNO groups showed significant downregulation of PPAR-γ expression in epididymal fat tissue relative to HFD group, the expression of PPAR-γ was upregulated in the HFD group compared to the normal diet (ND) group [46]. PNO (822 and 1644 mg/kg) reduced the upregulation of epididymal fat leptin mRNA compared with the HFD group in a dose-dependent manner. Furthermore, PNO also downregulated lipoprotein lipase (LPL) mRNA expression. 

In chronic calorie restriction (CR), SIRTs, a class 3 NAD+ dependent deacetylase, are known to be increased. SIRTs down-regulate the expression of genes that promote atherosclerosis [6,82,83]. SIRT3 is upregulated under CR conditions and plays an essential role in controlling mitochondrial function. SIRT3 is the primary mitochondrial form that supports FA oxidation, strengthens the antioxidant defense system, and repairs mitochondrial DNA damage [82]. In HFD-induced obesity, PNO consumption reduced body fat and hepatic TG accumulation [49]. SIRT3 is a new target for protection against lipotoxicity and obesity-related metabolic problems [82]. SIRT3 gene expression was upregulated by 25 μM PNLA in purified monocytes from RA patients [6]. SIRT3 protein expression was up-regulated in the white adipose tissue of PNO-fed mice [34]. In particular, NF-κB RelA/p65, AP-1 family transcription factor c-Jun, and c-Myc are frequent SIRT1 substrates. SIRT1 deacetylated RelA/p65 and decreased NF-κB activity [84,85,86]. SIRT1 significantly reduced the production of IL-12 in human dendritic cells (DC) through its impact on the NF-κB transcription factor c-Rel [84,85]. In addition, other studies have shown that SIRT1 in chondrocytes and synovial cells regulates various cell types during inflammatory arthritis [84,85,87]. Reports also confirmed that LPS-induced levels of TNFα in monocytes were decreased by overexpression of an inactive form of SIRT1 [87].

## 8. PNLA and Modulation of miRNAs

Takala et al., for the first time, showed a novel correlation between the relative expression levels of the miRNA’s targets and PNLA treatment. The miRNAs that were up or downregulated from 8 RA patients’ pure monocytes with active disease are summarized in Table 5. Monocytes upon PNLA treatment and LPS stimulation versus vehicle treatment and LPS stimulation resulted in differential expression of 68 miRNAs, 8 down-regulated and 60 up-regulated [6].

Table 6 displays the pathways connected to inflammation and lipid/cell metabolism that PNLA may control. The targets of these miRNAs were discovered through bioinformatic analyses to be several metabolic and anti-inflammatory mRNAs, including pyruvate dehydrogenase kinase-4 (PDK4), single-immunoglobulin interleukin-1 receptor-related molecule (SIGIRR), mitochondrial-ATP-6 (MT-ATP6), ghrelin and obestatin prepro-peptide (GHRL), jumonji domain containing 7-phospholipase A2 (JMJD7-PLA2G4B), and electron transfer flavoprotein subunit alpha (EFTA). These genes play a role in the control of inflammation, mitochondrial dysfunction, and lipid and cell metabolism. Adipocyte function, glucose metabolism, insulin secretion signalling pathways, CCK secretion, mitochondrial ATP synthase, the NAD signalling pathway, and FA β-oxidation were among the metabolic signalling pathways identified.

Ten miRNAs target PDK4, which controls TNF-α and NF-κB [5,88]; miR-3173, miR-2861, miR-626, miR-28, miR-7150, miR-3188, miR-879-5p, miR-393-5p, miR-708-5p, and miR-12118 [6]. Cellular metabolism depends on PDK4, and PDK4 deficiencies result in the death of hepatocytes by apoptosis with an increase in mitochondrial numbers, ROS production, prolonged activation of c-Jun N-terminal kinase (JNK), and decreased glutathione levels. Intracellularly, PDK4 can interact with proteins with death domains such as NF-κB and retain them in the cytoplasm [88]. According to the IPA database, three miRNAs, miR-7150, miR-1909-3p, and miR-6868-5p, target SIGIRR [6], which is a downstream regulator of TNF-α, IL-1β, TRIM1 and NF-κB signalling pathways. SIGIRR blocks IL-1R and TLR signalling by interacting with interleukin-1 receptor-associated kinase 1(IRAK-1), a protein linked with the interleukin-1 receptor, and TNF receptor-associated factor 6 (TRAF-6) [89]. Overexpression of SIGIRR reduced TLR-induced production of IL-10, TNF-α, and IFN-γ-induced protein 10 (IP-10) in DCs and HMDMs. Mice lacking SIGIRR experienced more severe collagen-induced arthritis (CIA) [89]. The suppression of naturally produced cytokines was caused by overexpression of SIGIRR in human RA synoviocytes [89]. MT-ATP6 is essential for mitochondrial function, sirtuin signalling, OXPHOS, and glucocorticoid receptor signalling [6]. Let7s are a family of miRNAs that have been shown to have established anti-atherogenic and anti-inflammatory effects. These effects include the control of smooth muscle cells and vascular endothelial cells, both of which are crucial for the emergence of atherosclerosis [74,75]. VSMC migration and proliferation is influenced by Let7 negative feedback control [74].

## 9. Discussion and Conclusions 

Studies indicate that PNLA has potential benefits as a dietary supplement or complementary therapy for chronic inflammatory and immune diseases as well as metabolic disorders. Anti-inflammatory and lipid-lowering drugs (e.g., immunosuppressives, fibrates, statins, bile acid sequestrants, etc.) reduce disease symptoms and reduce lipid levels through various mechanisms but often have adverse side effects. Additionally, considerable residual activity for the disease is often associated with such drugs. Therefore, additional research and clinical trials are required to reinforce PNLA’s anti-inflammatory effects to be included in medical treatment for patients. For example, studies on the role of PNLA on atherosclerosis could include mouse models of the disease or associated risk factors [90,91,92]. Some of the differences in findings on PNLA in the literature could represent different sensitivities of various cell types to the FA, or they could be a result of variation in experimental designs (e.g., inflammatory stimulant, FA concentration, incubation time). However, when considered collectively, current research and older studies show that PNLA may be capable of reducing the production of a plethora of pro-inflammatory cytokines, and oxidative mediators [5,19,27]. Data suggest that PNO and PNLA lower plasma lipids, including both cholesterol and TAG [2,67]. Pine nuts also include a variety of non-PNLA compounds that may have anti-inflammatory and metabolic effects. More research is needed to explore their potential. The unique UPIFA structure in PNLA might prevent it or its major metabolite (ETA) from being metabolised into PGE2. The latter at a lower concentration may reduce COX-2 expression, which as a result decreases the production of MMP in colon cancer cells [93], which further supports the anti-inflammatory effect of PNLA. PGE2 leads to inflammatory pain, and the reduction of PGE2 was also confirmed by Takala et al. in samples from HCs and RA patients [5].

In HepG2 cells and mouse livers, PNLA and pine nuts regulate lipoprotein uptake and reduce the expression of genes involved in FA biosynthesis [25,67]. Furthermore, PNO has several actions that lead to enhanced energy expenditure by increased oxidative metabolism via brown adipose tissue thermogenesis and decreased appetite through increased production of anti-satiety hormones. Reduced adipose tissue deposition, weight gain, and ectopic fat deposition all could result from these effects. Accordingly, PNO increased the expression of genes related to FA oxidation, mitochondrial oxidation, and oxidative metabolism in skeletal muscle [4]. This may explain the ability of PNO and PNLA to regulate muscle and cellular OXPHOS and metabolism.

There is little research on humans because many of the experiments covered in this review were carried out on cell lines or in animal models. Additionally, studies have been of short duration, so the long-term impact has not been established. High-quality clinical trials in humans for both PNO and PNLA will be necessary to establish the optimal doses and their effects on health and diseases. In addition, a well-known, important gut hormone glucagon-like peptide-1 (GLP-1) controls appetites, body weight, and glucose metabolism, which are controlled by PNO and PNLA. It is therefore important to determine whether formulation such as hydrolysed PNO in slow-release capsules may have additional positive impacts specifically in individuals with prediabetes or overt 2DM, in well-powered short- and long-term trials.

## 10. Prospects and Future Directions

Even though PNLA has biological activity, not all PNO’s effects might be attributable to it. As previously mentioned, pine nuts and PNO include a variety of additional, frequently at modest levels, molecules with potential biological activity, such as phytosterols, tocopherols, and squalene. It will be crucial to distinguish the effects of PNLA from those of the other components of PNO and not to ignore the potential contribution of these components to any biological effect identified for PNO. To assess their potential more accurately, it will be fundamental and worthwhile to examine how this FA affects the metabolism and mediates its anti-inflammatory effects. One such aspect is the lack of evaluation of PNLA’s effects on synovial histology and radiographic progression, as well as their function in early arthritis and their interaction with biologic medications.

## Figures and Tables

**Figure 1 ijms-24-01171-f001:**
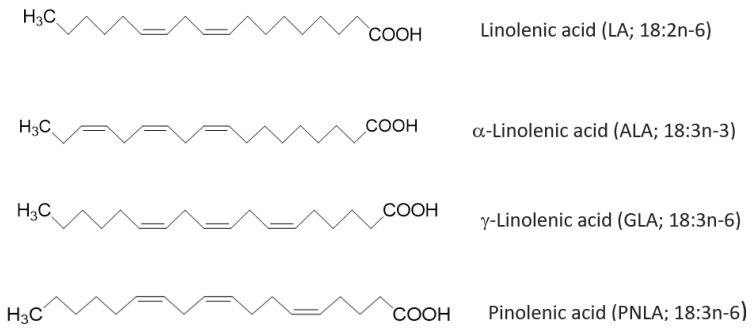
The simple linear structure of polyunsaturated fatty acids. Linolenic (LA), α-linolenic (ALA), γ-linolenic (GLA) and pinolenic acid (PNLA) are shown together with similarities between these FAs and the locations of double bonds.

**Figure 2 ijms-24-01171-f002:**
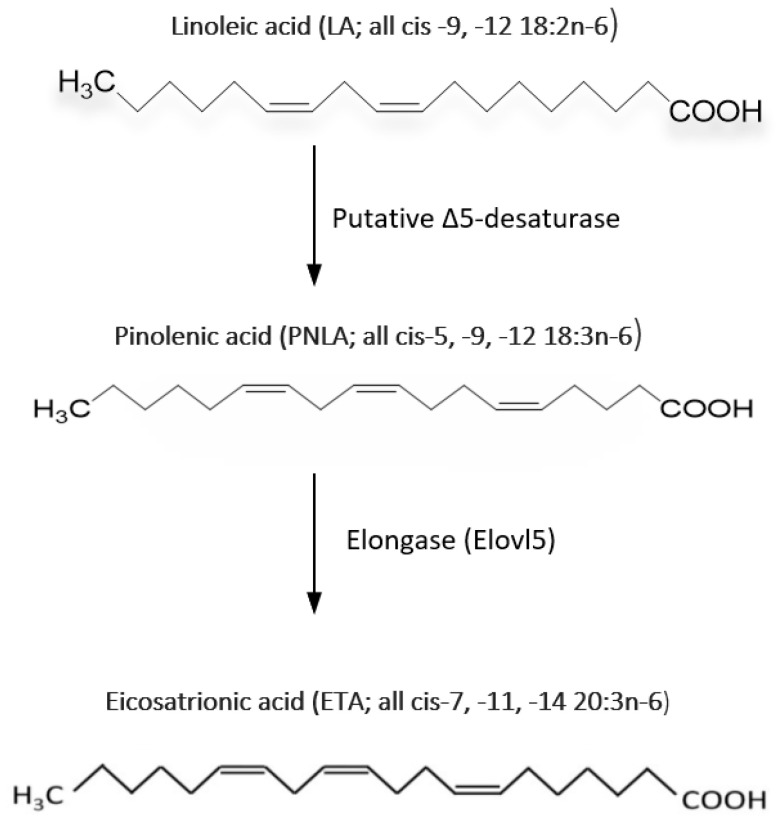
The pathway of diversion of linoleic acid to pinolenic and eicosatrienoic acids. Δ5-desaturase enzyme deoxidises LA into PNLA. The ELOVL5 gene encodes the elongase 5 enzyme and is involved in PNLA elongation into ETA. ELOVL5; Elongase of Very-Long Fatty Acid 5.

**Figure 3 ijms-24-01171-f003:**
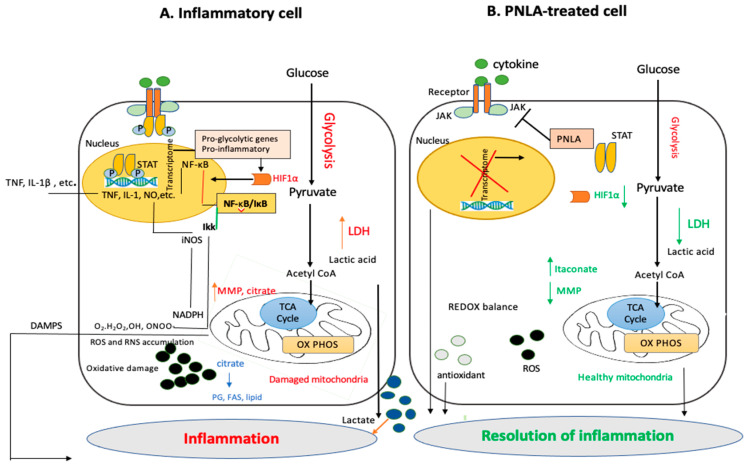
Summary on the consequences of oxidative phosphorylation (OXPHOS) in inflammatory and PNLA treated cells. (**A**) Metabolic pathway in inflammatory cell models. Glucose enters the cell and is converted into pyruvate via glycolysis, which is then converted to acetyl CoA that then moves into the mitochondria to undergo OXPHOS under inflammatory conditions. MMP and citrate are produced; citrate is a source of PGs (e.g., arachidonic acids) and ROS such as O_2_−, which binds with peroxy-nitrite (ONOO-) to form RNS. This reactive species plays a role in the NF-κB-mediated production of inflammatory mediators, such as TNF, IL-1β and NO. The synthesis of NO is mediated by the enzyme iNOS. Additionally, this can lead to the activation of MMP and depletion of GSH, all of which lead to oxidative damage. Under inflammatory conditions, HIF-1α is also produced which activates pro-inflammatory genes such as NF-κB and STAT. JAKs phosphorylate STATs, and once phosphorylated, this leads to the transcription of pro-inflammatory cytokines. (**B**) Potential actions of PNLA. PNLA reduces LDH via activation of PDK4, which leads to less lactate and acetyl CoA that consequently results in the inhibition of OXPHOS. PNLA treatment can produce less HIF1-α, and oxidative radicles; this together with more itaconate produced results in redox balance hence the resolution of inflammation. Additionally, PNLA inhibits the activation of NF-κB and STAT leading to the inactivation of transcription of pro-inflammatory cytokines. FAS, fatty acid synthase; HIF-α, hypoxia-inducible factor 1; H_2_O_2_, hydrogen peroxide; LDH, lactate dehydrogenase; iNOS, inducible nitric oxide synthetase; JAK, Janus kinase; MMP, matrix metalloproteinase; NO, nitric oxide; OXPHOS, Oxidative phosphorylation; PDK-4, pyruvate dehydrogenase kinase-4; PG, prostaglandin; ROS, reactive oxygen species; RNS, reactive nitrogen species; STAT, signal transducers and activators of transcription; TCA cycle, tricarboxylic acid cycle; O2−, superoxide.

**Table 1 ijms-24-01171-t001:** A summary of the anti-inflammatory effects of pine nut oil (PNO) or pinolenic acid (PNLA).

Model	PNO or PNLA	Experimental Design	Results and Outcome of PNO and PNLA Treatment	Reference(s)
Murine RAW264.7 macrophages	PNLA	10, 25, 50 or 100 μM PNLA for 24 h followed by lipopolysaccharide (LPS) stimulation (100 ng/mL) for 16 h. 50 μM PNLA for 24 h followed by LPS stimulation (100 ng/mL) for 16 h.	Release of PGE1 by RAW264.7 cells is reduced by 10, 25, 50 and 100 μM PNLA by 33, 49, 73, and 84%, respectively. COX-2 is not affected by 50 μM PNLA.	[16]
Human breast cancer MDA- MB-231 cells	PNLA	50 or 100 μM PNLA for 24 h followed by 12-*O*-tetradecanoylphorbol-13-acetate (TPA) stimulation (100 ng/mL) for 12 h. 100 μM PNLA for 24 h followed by TPA stimulation (0.1 μg/mL) for 12 h.	Release of PGE2 is reduced by 50 and 75% by 50 and 100 μM PNLA, respectively. COX-2 mRNA and protein is both reduced about 55% by 100 μM PNLA.	[22]
Murine macrophage RAW264.7 cells and rat primary peritoneal macrophages	PNLA	25, 50 and 100 μM PNLA for 24 h followed by LPS stimulation (100 ng/mL) for 16 h. 50 and 100 μM PNLA for 24 h followed by LPS stimulation (0.1 μg/mL) for 8 h. 50 and 100 uM PNLA for 24 h followed by LPS stimulation (100 ng/mL) for 30 min. 50 and 100 μM PNLA for 24 h followed by LPS stimulation (100 ng/mL) for 15 min.	Release of PGE2 by RAW264.7 cells at 50 and 100 μM PNLA is reduced by 67% and 80%, respectively. Release of PGE2 by perineal macrophages is reduced by 13% by 50 μM PNLA but unaffected by 25 μM PNLA. COX-2 in RAW264.7 cells is reduced by 50 and 100 μM PNLA by 20 and 40%, respectively. NF-κB/p65 protein ratio in RAW264.7 cells treated with 50 and 100 μM PNLA is reduced by 40 and 50%, respectively.	[29]
Murine microglial BV-2 cells and rat primary peritoneal macrophages	PNLA	50 μM PNLA for 24 h followed by LPS stimulation (100 ng/mL) for 16 h.	Release of IL-6, TNF-α, NO and PGE2 by BV-2 cells is reduced by 71, 27, 41 and 89%, respectively. In BV-2 cells, iNOS and COX-2 proteins expression are reduced by 53 and 10%, respectively. Production of NO and PGE2 by peritoneal macrophages is reduced by 31 and 35%, respectively.	[17]
THP-1 macrophages	PNLA	10, 25, 50 and 100 μM PNLA for 24 h followed by LPS stimulation (200 ng/mL) for 16 h.	Release of TNF-α is reduced by 9 and 18%, respectively by 50 and 100 μM PNLA. IL-6 is reduced by 9, 24, 33 and 48% by 10, 25, 50 and 100 μM PNLA, respectively. PGE2 is reduced by 55, 67, 78, and 83%, respectively by 10, 25, 50 and 100 μM PNLA. Protein expression of COX-2 is reduced by 50 and 100 μM PNLA by 20 and 25%, respectively.	[19]
Hep G2 cells	PNLA	25 μM PNLA for 12 h followed by 0.5 mM oleic acid for 24h.	Synthesis of NO by HepG2 cells is reduced by 60% with 25 μM PNLA.	[25]
EA. hy296 cells	PNLA	10, 25 and 50 μM PNLA for 48 h followed by TNF-α stimulation (1 ng/mL) for 24 h. 50 μM PNLA for 48 h followed by TNF-α stimulation (1 ng/mL) for 1 h. 25 and 50 μM PNLA for 48 h followed by TNF-α stimulation (1 ng/mL) for 6 h.	Levels of soluble intercellular adhesion molecule (ICAM)-1 by 10, 25, and 50 μM PNLA is reduced by 15, 23 and 24%, respectively. Release of monocyte chemotactic protein (MCP)-1 is reduced with 50 μM PNLA by 25%. Production of regulated on activation, normal T cell expressed and secreted (RANTES) is reduced by 46% with 50 μM PNLA. Phosphorylated-NFκB/NFκB protein ratio is reduced by 50%. Adherence of THP-1 cells to EA.hy296 cell monolayers is reduced by 25% with 50 μM PNLA.	[20,27]
THP-1 monocytes, PMA-differentiated THP-1 macrophages and human monocyte-derived macrophages (HMDMs).	PNLA	25, 50, 75 and 100 μM PNLA for 24 h followed by 20 ng/mL MCP-1 stimulation for 3 h in monocyte migration, and Lucifer Yellow (LY) and oxidized LDL (ox-LDL) incubation for 24 h for macropinocytosis and Dil-ox-LDL uptake, respectively for both THP-1 macrophages and HMDMs.	The mean decrease in MCP-1-mediated THP-1 monocyte migration across all PNLA concentrations was 55%. Macropinocytosis and DiI- ox-LDL uptake was reduced by 50% and 40% in THP-1 macrophages and by 40% and 25% in HMDM, respectively.	[5]
Peripheral blood mononuclear cells (PBMCs) from healthy controls (HCs) and rheumatoid arthritis (RA) patients.	PNLA	25 and 50 μM PNLA for 24 h followed by LPS stimulation (100 ng /mL) for 16 h.	TNF-α and IL-6 levels in cell free supernatants were reduced by 60% from RA patients and in HCs were reduced by 50 and 35%, respectively by 25 and 50 μM PNLA. PGE2 was reduced by 50% from both HCs and RA patients by 50 μM PNLA.	[5]
CD14 monocytes purified from RA patients with active disease.	PNLA	25 and 50 μM PNLA for 24 h followed by LPS stimulation (0.1 μg /mL) for 9 h. The proportions of CD14 monocytes, or CD14 monocytes expressing TNF-α, IL-6, IL-1β, and IL-8 in purified monocytes were assessed by flow cytometry.	The percentage of monocytes expressing TNF-α, IL-6, and IL-1 was decreased by 23%, 25%, and 23%, respectively, by PNLA. Percentages of CD14^+^ monocytes were reduced by 20% following 25 or 50 μM PNLA.	[6]
Male Wistar rats	PNO (*P. sibirica*)	300 mg/kg bodyweight PNO for 2 days and 4 h prior to carrageenan injection into right feet and exposing the feet to a heat of 55 °C.	Swelling (paw volume) at 3-, 12- and 24 h following carrageenan injection was reduced by 24, 36 and 45%, respectively. Fever-reducing effect (surface temperature of adjuvant-inflamed paw) was reduced by 5%, 10% and 10%, respectively following 3, 12 and 24 h after carrageenan injection. Analgesic effect (response time to 55 °C thermal-induced hot-plate) was induced by more than 100% following 3 and 12 h while unaffected following 24 h.	[24]
Male ICR mice	PNLA	PNLA (3 g) was injected intradermally into the ears for 18 hours, followed by TPA (5 g) for 6 or 24 h. Topical application of PNLA (3 μg) to dorsal skin followed by TPA injection (5 μg) for 2 h.	Ear swelling, thickness and COX-2 protein expression in mouse ear tissue homogenates was reduced by 29, 15, and 53%, respectively. Infiltration of leukocytes (CD45+), neutrophils (Ly6G+CD45+) and macrophages (F4/80+CD45+) was reduced by 63, 50 and 70%, respectively. IL-1, IL-6, TNF-α, and PGE2 levels in the dorsal skin cell-free supernatant was reduced by 79, 68, 42 and 51%, respectively. Phosphorylated p38 expression was dorsal skin tissue is reduced by 55%.	[19]

COX, cyclooxygenase; Dil-ox-LDL, DiI-labelled oxidized low-density lipoprotein; ERK, extracellular signal-regulated kinase; HMDM, human monocyte-derived macrophages; HC, healthy controls; hour, h; ICAM, intercellular adhesion molecule; IL, interleukin; iNOS, inducible nitric oxide synthase; JNK, c-Jun N-terminal kinase; LPS, lipopolysaccharide; LY, Lucifer yellow; MCP, monocyte chemotactic protein; NF-κB, nuclear factor kappa-light-chain-enhancer of activated B cells; NO, nitric oxide; PGE2, prostaglandin E2; PBMCs, peripheral blood mononuclear cells; PNLA, pinolenic acid; PMA, phorbol 12-myristate 13-acetate; RA, rheumatoid arthritis; RANTES, regulated on activation normal T cell expressed and secreted; TNF, tumour necrosis factor; TPA, 12-*O*-tetradecanoylphorbol-13-acetate.

**Table 2 ijms-24-01171-t002:** The top ten genes whose expression in LPS-stimulated PBMCs from HCs was affected by pre-treatment with PNLA compared to vehicle.

Gene Name	Gene Biotype	Change in Expression
PDK4	protein coding	increase
TMEM52B	protein coding	increase
AC092118.1	lncRNA	increase
CPT1A	protein coding	increase
SERPINE1	protein coding	increase
AC087289.4	lncRNA	increase
IGSF6	Protein coding	increase
AKR1C1	Protein coding	increase
AC138207.5	lncRNA	increase
TSPAN10	Protein coding	increase

**Table 3 ijms-24-01171-t003:** The top ten genes whose expression in LPS-stimulated PBMCs from RA patients was affected by pre-treatment with PNLA compared to vehicle.

Gene Name	Gene Biotype	Change in Expression
FBP1	protein coding	increase
PCAT7	lncRNA	increase
NDRG2	protein coding	increase
PDK4	protein coding	increase
AC015660.2	lncRNA	increase
LINC02244	lncRNA	increase
LRRC32	protein coding	increase
LOXL2	protein coding	increase
CD163L1	protein coding	decrease
DIXDC1	protein coding	increase

**Table 4 ijms-24-01171-t004:** Upregulated and downregulated protein-coding genes in CD14 monocytes from patients with RA following PNLA treatment and LPS stimulation versus vehicle treatment and LPS stimulation.

Upregulated Genes	Downregulated Genes
LY6G5B	SPCS1
PDK4 *	RHNO1
BRF1	MRPL9
ACAA2	MT-ND1
ZBTB34	HSPA1L
ACADVL *	MT-CO2
AC007375.2	CHCHD4
SPINK4	NIT1
AC090227.2	OTUB1
CPT1A *	PAIP1
GRIK1	MEN1
HSD17B8	OTUB1
NPEPL1	MEN1
ZNF48	MT-ATP6
ANKRD23	ENOX2
CCER2	LSM1
SLC25A42	ARL2BP
GHRL	MT-ND5
ALG13	MT-ND4
CRABP2	SLC10A3
MTRNR2L8	DEDD
ROM1	DSTN
ST14	DCTN2
SIGIRR	TRAPPC2L
ETFA	EDARADD
JMJD7-PLA2G4B	NDUFA7
SLC25A20 *	GZF1

* Indicate genes that are consistently regulated by both CD14 monocytes and PBMCs of RA patients in separate studies.

**Table 5 ijms-24-01171-t005:** Key down-regulated and up-regulated miRNAs in PNLA treated, LPS-stimulated versus vehicle treated LPS-stimulated CD14 monocytes from patients with RA. “Adapted with permission from Takala et al. [6]. Copyright 2022, Scientific Reports”.

Downregulated miRNA	Upregulated miRNA
mIR637	mIR8066
mIR4326	mIR1276
mIR6886	mIR3173
mIR1909	mIR664B
mIR671	mIR6773
mIR7111	mIR6778
	mIR374C
	mIR374B
	mIR3161
	mIR219B
	mIR3922
	mIR219A2
	mIR1914
	mIR505
	mIR3140
	mIR941-3
	mIR324
	mIR4722
	mIR4755
	mIR3176
	mIR3978
	mIR4435-2
	mIR4440
	mIR1470
	mIR570
	mIR6719

**Table 6 ijms-24-01171-t006:** Selected miRNAs target specific mRNAs and the signalling pathways that they are linked to, according to the Ingenuity pathway Analysis (IPA) database. “Adapted with permission from Takala et al. [6]. Copyright 2022, Scientific Reports”.

miRNA ID	Change in Expression	mRNA Target	Effect	Pathway(s)
miR-3173	increase	CRABP2	Increase	Acute phase response, signalling for apoptosis mediated by retinoic acid, and regulation of cellular processes by glucocorticoids.
miR1260B	increase	JMJD7-PLA2G4B	Increase	VEGF family ligand-receptor interactions, ERK/MAPK signalling, glucocorticoid receptor signalling, MIF regulation of innate immunity, MIF-mediated glucocorticoid regulation, p38 MAPK signalling, and phospholipase C signalling.
miR-646	increase			
miR-1909	decrease	FZD2	Decrease	Pathways for adipogenesis, osteoarthritis, and the control of macrophages, fibroblasts, and endothelial cells in rheumatoid arthritis, as well as osteoblasts, osteoclasts, and chondrocytes.
miR-1909	decrease	SIGIRR	Increase	Signalling from the NF-κB transcription factor, TLRs, and TREM1. Signalling by IL-6, TNF-α, and IL-1. Both anti-atherogenic and anti-inflammatory pathways.
miR-7150	increase			
miR-6868-5P	increase			
miR-2861	decrease	LSM1	Decrease	Systemic lupus erythematosus signalling.
miR-374B	increase	ETFA	Increase	NAD signalling pathway involved in energy production from fats and proteins.
miR-4440	increase	ATMIN	Decrease	Control of the cell cycle checkpoint by CHK proteins. Role in development of the immune system.
miR-4440	increase	GHRL	Increase	Leptin signalling in obesity, appetite and growth hormone regulation.
miR-548L	increase			
miR-626	increase	PDK4	Increase	Reelin signalling, glucocorticoid receptor signalling, and senescence pathways. TNF-α and NF-κB checkpoint. Lipid and glucose metabolism regulation. Associated with mitochondrial function and cellular energy regulation. Improves VSMCs oxidative stress resistance.
miR-3173	increase			
miR-2861	decrease			
miR-28	increase			
miR-7150	increase			
miR-3188	increase			
miR-637	Decrease	SPCS1	Decrease	Insulin secretion signalling pathways.
miR-671	Decrease	MT-ATP6	Decrease	Glucocorticoid receptor signalling, mitochondrial dysfunction, oxidative phosphorylation, sirtuin signalling pathway.
miR-Let-7	increase	PDE12 RRP1B TARBP-2 GZF1 NRAS	All decrease	Anti-inflammatory and anti-atherogenic actions. Suppression of immune-modulatory cytokines IL-6 and IL-10. TLR4 signalling.

CRABP2, Cellular Retinoic Acid Binding Protein; CHK, Checkpoint Kinase; ETFA, Electron Transfer Flavoprotein Subunit alpha; ERK/MAPK, Extracellular signal-regulated kinase/Mitogen activated protein kinase; FZD2, Frizzled Class Receptor 2; GHRL, Ghrelin Additionally, Obestatin Prepropeptide; JMJD7-PLA2G4B, Jumonji Domain Containing 7-Phospholipase A2; LSM1, Like Protein U6 Small Nuclear RNA Associated; MIF, Macrophage migration inhibitory Factor; MT-ATP6, mitochondrial encoded ATP synthase membrane subunit-6; NAD, Nicotinamide adenine dinucleotide; PDK4, pyruvate dehydrogenase kinase-4; PDE12, phosphodiesterase-12; SPCS1,Signal Peptidase Complex Subunit 1; TLR, Toll-like receptor; NF-κB; nuclear factor kappa-light-chain-enhancer of activated B cells; SIGIRR, single-immunoglobulin interleukin-1 receptor-related molecule; TREM1, Triggering receptor expressed on myeloid cells 1; VSMC, Vascular smooth muscle cells; VEGF, vascular endothelial growth factor.

## Data Availability

Not applicable.

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
