# Peer review of "The Beneficial Effects of Pine Nuts and Its Major Fatty Acid, Pinolenic Acid, on Inflammation and Metabolic Perturbations in Inflammatory Disorders"

_ijms, 2023, doi:10.3390/ijms24021171_

Round 1

Reviewer 1 Report

Very interestin review.

I have several suggestion:

1.Line 42; citation is missing

2.Line 45; Correct the sentence it is not clear.

3. Fig. 1. should be improved.

4. Table 4. A and for B should be connected and present as one table.

5.Table 5 A and 5 B , the same.

Author Response

Point 1. Line 42; citation is missing

Response 1. The citation has been added [1]

Point 2. Line 45; Correct the sentence it is not clear.

Response 2. The sentence has been modified and made clearer.

Point 3. Fig. 1. should be improved.

Response 3. The figure is more organized and clearer now.

Point 4. Table 4 A and B should be connected and present as one table.

Response 4. Tables 4 A and B have been combined into one table (Table4).

Point 5. Table 5 A and 5 B are the same.

Response 5. Tables 5A, and B have been merged into one table (Table5).

Reviewer 2 Report

The article written by Takala and collaborators is a very interesting article that analyses the benefits of using PNO and PNLA in inflammatory and metabolic disorders, including gene expression analysis, summarizing some relevant articles on this topic.

However, the authors did not respect the instructions for presenting the references. References must be numbered in order of appearance in the text, and reference numbers should be placed in square brackets [ ].

Also, the list of references does not respect the Instructions for authors.

The authors refer in the text to the articles by Baker and his collaborators from 2018, 2020 and 2021, but there is only one article in the list of references:

Baker EJ, Valenzuela CA, van Dooremalen W, Martínez-Fern ́andez L, Yaqoob P, Miles EA, et al. 2020.Gamma-linolenic and 884 pinolenic acids exert anti-inflammatory effects in cultured human endothelial cells through their elongation products. Mol Nutr 885 Food Res;64:e2000382.

A very important article (Baker EJ, Miles EA, Calder PC. A review of the functional effects of pine nut oil, pinolenic acid and its derivative eicosatrienoic acid and their potential health benefits. Prog Lipid Res. 2021 Apr;82:101097. doi: 10.1016/j.plipres.2021.101097. Epub 2021 Apr 5. PMID: 33831456) is missing from the References list; it has many points in common with the present article by Takala and collaborators.

On the other hand, the tables do not have a uniform format and give the appearance of an unfinished article.

The authors seem to have not studied as many recent articles as it appears from the list of references and they have comprehensively presented their own results.

Although [Maxwell 2022] (line 358) is cited in the text, I did not find this article in the References list

I believe that the article must be reviewed, modified and completed in order to be published.

Author Response

Point 1. The article written by Takala and collaborators is a very interesting article that analyses the benefits of using PNO and PNLA in inflammatory and metabolic disorders, including gene expression analysis, summarizing some relevant articles on this topic. However, the authors did not respect the instructions for presenting the references. References must be numbered in order of appearance in the text, and reference numbers should be placed in square brackets [ ]. Also, the list of references does not respect the Instructions for authors. 

Response 1. All the references now have been numbered in square brackets [ ], in order following the flow of the text, the citations are all in red now to be easier to follow.

Point 2: The authors refer in the text to the articles by Baker and his collaborators from 2018, 2020 and 2021, but there is only one article in the list of references:

Response2. Apologies for this, Baker et al references are now all on the list numbered [20] [21] [27].

Point 3: On the other hand, the tables do not have a uniform format and give the appearance of an unfinished article.

Response3. All the tables are now organized and structured in one format.

Point 4. The authors seem to have not studied as many recent articles as it appears from the list of references, and they have comprehensively presented their own results. Although [Maxwell 2022] (line 358) is cited in the text, I did not find this article in the References list.

Response 4. We have tried to focus the review article on animal and human studies especially as reviews on other aspects exist. Maxwell in 2022 gave a very interesting talk at Cardiff University regards the therapeutic advances in HIF-1 inhibition. However, it seems the article has not been published yet. However, looking at the same theme in the literature lots of articles were mentioned this. I added [39] by Li et al 2020. HIF-1α is a potential molecular target for herbal medicine to treat diseases. Drug design, development, and therapy, 14, 4915.‏

Round 2

Reviewer 2 Report

I appreciate that the authors took into account my suggestions, and corrected and completed the article so that it could be published.